# Catalytic Reduction of Graphene Oxide Membranes and Water Selective Channel Formation in Water–Alcohol Separations

**DOI:** 10.3390/membranes11050317

**Published:** 2021-04-26

**Authors:** Yushi Zang, Alex Peek, Yongsoon Shin, David Gotthold, Bruce J. Hinds

**Affiliations:** 1Department of Material Science and Engineering, University of Washington, Seattle, WA 98105, USA; yushiz@uw.edu (Y.Z.); peek@uw.edu (A.P.); 2Pacific Northwest National Laboratory, Richland, WA 99352, USA; yongsoon.shin@pnnl.gov (Y.S.); david.gotthold@pnnl.gov (D.G.)

**Keywords:** catalytic membranes, nanofluidics, biofuel separations

## Abstract

Graphene oxide (GO) is a promising membrane system for chemical separation applications due to its 2-D nanofluidics properties and an ability to control interplanar spacing for selectivity. The permeance of water, methanol (MeOH) and isopropyl alcohol (IPA) through 5 µm thick membranes was found to be 0.38 ± 0.15, 0.33 ± 0.16 and 0.42 ± 0.31 LMH/bar (liter/m^2^·h·bar), respectively. Interestingly, the permeance of a water–alcohol mixture was found to be dramatically lower (~0.01 LMH/bar) than any of its components. Upon removing the solvent mixture, the transmembrane flux of the pure solvent was recovered to near the original permeance. The interlayer space of a dried GO membrane was found to be 8.52 Å, which increased to 12.19 Å. 13.26 Å and 16.20 Å upon addition of water, MeOH and IPA. A decrease in d-space, about 2 Å, was consistently observed when adding alcohol to water wetted GO membrane and an optical color change and reduction in permeance. A newly proposed mechanism of a partial reduction of GO through a catalytic reaction with the water–alcohol mixture is consistent with experimental observations.

## 1. Introduction

Graphene oxide (GO) is a promising membrane system for applications in chemical separations due to enhanced 2-D nanofluidics properties and an ability to control interplanar spacing for size-based chemical exclusion. Dramatic flow rate enhancements of 3–4 orders of magnitude have been experimentally seen through carbon nanotube (CNT) membranes with the atomically smooth surface of the graphitic planes making the tube walls [1,2]. Molecular dynamics simulations show that in addition to atomic smoothness [3], water molecules can be structured to a single-chain conformation connected by two hydrogen bonds, giving flow rates greater than aquaporin water channels [4]. A similar phenomenon was also observed in 2-D atomically flat channels of graphene with near-frictionless fast transport of liquid molecule [5,6] and, similarly, large slip length for graphene nanochannels [5,7]. Water molecules confined in 2-D channels of graphene were also observed to have an ordered structure different from bulk water, including “square ice” 2-D ordering [8,9,10]. Simulations of H-bonding of water within GO membranes predict slow water diffusion [11], adding complexity to this system for determining the dominant transport mechanisms. Additionally, nano-confinement effects give selectivity towards different chemical species [6,9,12,13,14]. However, the fabrication of mechanically robust membranes with 2-D channels and high porosity over large surface areas remains a challenge and is required for systematic chemical transport and selectivity studies.

Graphene oxide is produced by an industrially scalable process of chemically oxidizing bulk graphite, which is readily dispersed in aqueous solutions, thereby allowing for membrane fabrication by simple filtration. The hydrophilicity of GO is due to oxidation product moieties, such as hydroxyl, carbonyl, carboxyl, and epoxide groups covering roughly 18–36% of total carbon sites [15]. It is important to note that such oxygen-containing moieties are not evenly distributed across the GO surface and play a complex role in determining the properties of GO nanochannels [15,16]. For instance, stronger interaction between water molecules and oxidized region of GO can cause greater hindrance of trans-membrane flux [10,17,18,19]. Others proposed a mechanism in which the oxygen-containing groups on the entrance of GO nanochannels disrupt the hydrogen bonding of bulk water, thereby lowering entrance enthalpy and reducing flow [20]. Although the true flow dynamic inside the GO nanochannel is yet to be revealed, an approximation where the fast water flow was allowed by pristine graphene region formed “frictionless channels” was proposed with good agreement with experiment permeance observations [18,21]. Aside from the chemical condition of graphene oxide sheet, the separation performance of a GO membrane are also greatly affected by its fabrication process. Common GO membrane fabrication processes include direct drop-casting [22], filtering [9,23], slip-casting [24,25] of GO dispersion, and mixing the GO flakes into another matrix but have widely varying reported permeances. Permeability is defined as “membrane thickness × permeance of solvent” and is more appropriate for comparison of membranes produced by different methods resulting in significantly different thicknesses [12,26]. The membrane community commonly uses a permeance nomenclature of LMH/bar, which is liters of permeate per m^2^ of membrane area per hour per bar pressure applied. For instance, one study showed that pressure-assisted filtered GO membrane could have a permeance (permeability) performance of about 2.5 kg/m^2^·h (0.59 µm × liter/(m^2^·h·bar) or µm × LMH/bar) with a water–alcohol (1:9) mixture, while the drop-casted membranes only show 0.9 kg/m^2^·h (0.41 µm × LMH/bar). A vacuum-assisted filtered membrane shows 2.4 kg/m^2^·h (0.91 µm × LMH/bar) under the same condition, demonstrating the importance of the membrane synthesis methods to give ordered laminate nanochannel structures [23]. The slip-casting process was systematically optimized for GO membrane fabrication to improve the microstructure by shear-aligning GO flakes. Such membranes were found with water permeance as high as 71 LMH/bar (10.65 µm × LMH/bar), which is over 7 times the permeance of common filtered GO membranes [24]. The fabrication processes mentioned above also influence the interlayer space, or d-space, of the GO membrane, normally about 0.8 nm at dry state [23,25]. Due to high hydrophilicity, the membrane can expand as large as 6 nm [17] and even redisperse into a suspension when wetted with water over an extended time [15]. Cations stabilize the GO membrane through electrostatic screening of anions of carboxyl groups of adjacent GO sheets and can tune the d-spacing [27]. In general, GO membranes require ~10 mM salt concentration in water to prevent the membrane’s dissolution, limiting its applications. Forming chemical bonds between neighboring GO sheets with a designed crosslinking chemical is another elegant way to control the d-space of GO [28,29,30]. Larger spacers, such as engineered nanoparticles, carbon nanotubes, and polymers, can be intercalated in GO membranes to promote higher trans-membrane flux with larger interlayer spacing [26,31,32,33,34].

Energy-efficient water–alcohol separation is an unmet technical challenge for biomass-derived sustainable energy fuels. Particularly challenging is the tight control of channel size to give water selectivity over large membrane areas. Recent advances in slip-casting have led to using an optimized shear rate to align the GO grains in 5 µm thick GO membranes for ethanol dehydration via pervaporation [25]. Studies have purposed a concept of critical thickness, which is the required GO membrane thickness to avoid defect like pinholes that goes across the membrane, with 5 µm being a usable thickness [12,13,22]. Interestingly, in the pervaporation study, fluxes dropped dramatically for water–alcohol mixture, but not for individual components [25]. A similar flux decrease phenomenon was also observed in an attempt to separate water from ethanol and isopropyl alcohol [22,35]. Proposed mechanisms include forming a water–GO complex at the GO channel entrance that reduces the effective hydrodynamic pore size of GO, excluding alcohol [25,35]. Others propose that alcohol molecules disrupt the entrance of water molecules into the GO plane [19,36,37,38].

Aside from applying chemical separation, GO was also considered a strong candidate for catalyzing a wide range of reactions. As a solid catalyst, free-standing GO powder was demonstrated to convert benzylic alcohols to corresponding aldehyde and ketone at the cost of GO being partially reduced [39]. Other reactions, such as ring-opening of epoxides, can also be catalyzed by GO with a good conversion rate and product selectivity [40]. GO was also made into membranes for efficient catalyst recovery after the reaction. Another GO-based membrane system, consisting of nitrogen-doped GO synthesized via chemical vapor deposition of NH_3_ and CH_4_ gas, demonstrated catalytic activity toward oxygen reduction in alkaline fuel cells for the first time [41]. Applications for GO-based catalytic membrane in other fields, such as biodiesel production and organic pollutants degradation, have also been developed [42,43].

Reported here is a study of pressure-driven liquid mass transport behavior through GO membranes fabricated by a carefully engineered slip-casting process. The permeance (permeability) of single solvents of H_2_O, methanol (MeOH) and isopropyl alcohol (IPA) (all with 10 mM, mmol/L, NaCl stabilizer) were 0.38 ± 0.15, 0.33 ± 0.16 and 0.42 ± 0.31 LMH/bar (1.90 ± 0.77, 1.65 ± 0.8 and 2.09 ± 1.54 µm × LMH/bar), respectively. However, after wetting the membrane with a water–alcohol mixture, the permeating flux reduced by more than tenfold. A similar flux decrease phenomenon was also observed in osmotic pressure-driven fluxes when a water–alcohol mixture was present on the feed side of the membrane. The osmotic flux can be recovered when using single component solvents. Osmosis experiments demonstrated rejection of salt and the selectivity of water over alcohol through GO membranes. This occurred with membrane color change to dark black and reduction in interplanar d-spacing by 0.2 nm. A newly proposed GO reduction mechanism by alcohol mediated by water is consistent with experiment and literature observations [15,18,44,45]. The reduction process of the GO membrane can be reversed with the addition of oxidant to restore permeance.

## 2. Materials and Methods

***Units:*** The units used in this study include: L for liter; mL for milliliter; g for gram; mg for milligram; rpm for revolutions per minute; h for hour; mol for mole; M for mol/L; mM for millimol/L; mm for millimeter; µm for micrometer; nm for nanometer; °C for degree celsius; LMH for L/m^2^·h; LMH/bar for L/m^2^·h·bar.

***Chemicals:*** Chemicals used for this study were obtained from Sigma-Aldrich (St. Louis MO USA) and used without further process unless otherwise stated.

***GO membrane preparation***: The GO membrane was prepared by PNNL following a previous report [25]. An amount of 5.0 g graphite powder (Asbury, NJ USA cat #3763, 500 µm flakes) was added to previously mixed H_2_SO_4_ (200 mL) and H_3_PO_4_ (40 mL). Then, 24.0 g of KMnO_4_ was slowly added to the mixture while constantly stirring, followed by 5 h of reaction. The mixture was then poured into 1.0 L of ice water with 5.0 mL of H_2_O_2_ (30 wt.%) to remove KMnO_4_ residue. After centrifuging the suspension for 5 min at 4000 rpm, the residual solid was then added to 1 M (mol/L) H_2_SO_4_ solution to remove the excess metal ion. The yielded mixture was again centrifuged for 10 min at 4000 rpm, followed by decanting the supernatant. The residual was continuously washed by deionized (DI) water and centrifuged at 9000 rpm for 30 min till a two-layer structure was formed, while the top layer was jelly-like and containing mostly exfoliated GO sheets. The top layer was collected and added to 1 L for centrifuge at 4000 rpm for 3 min to remove residue unexfoliated GO particles. Then, the remaining GO suspension was centrifuged to concentrate the exfoliated GO flakes. A brown slurry of GO suspension containing about 1 wt.% of GO was obtained by centrifuging at 10,000 rpm for 40 min and decanting the supernatant. Obtained GO slurry was poured onto a PES supporting membrane and slip-casted by a glass rod, then subsequently dried. It is important to physically secure the GO membrane for it to maintain its structure under pressure. Thus, supporting structures were employed. A 1.59 mm (1/16 inch) thick Delrin sheet was laser cut into 25.4 mm × 25.4 mm squares with an 8 mm circular opening in the center as support for the GO membrane. 400 mesh stainless steel mesh from Ted Pella, Inc. was used to provide support as well. Medical-grade Loctite M31-CL epoxy was used to glue the GO membrane and form liquid-tight seals to the supports according to the sequence in Figure 1.

***Chemical modification:*** Surface modifications were made to the GO membrane to accelerate the permeation of water. Chemical treatments of GO membranes were performed in a u-tube setup (Figure 1B) adapted from a previous study [46], except the treatment with concentrated sulfuric acid. After the GO membrane was secured between tubes, DI water was filled into the long u-tube as feed solution, while the chemical solution (0.076 mM, mmol/L, for the two diazonium chemistries as described below) was filled into the other tube (draw tube). The liquid level in the feed tube was intentionally kept higher than the drawtube to induce a cross membrane flux towards the drawtube. Therefore, the chemical treatment was limited to only one side of the membrane. Such treatments were maintained overnight in a dark cabinet due to the light sensitivity of diazonium.

For sulfuric acid treatment, the GO membrane was secured between a glass slide and a glass tube by spring-loaded clamp and tub filled (~0.5 mL) with concentrated sulfuric acid for 60 s followed by a rinse with DI water.

***Aryl Diazonium Salt Synthesis****:* The diazonium synthesis process used in this study were modified procedures of the diazotization process accommodating the nature of chemicals used.

***p-carboxyl terminated*:** An amount of 0.02 mol 4-amino benzoic acid was added into 20 mL of water at 50 °C while stirring, followed by the addition of 0.044 mol of concentrated hydrochloric acid. The mixture was cooled to −3 °C. After cooling, a solution of 0.022 mol of NaNO_2_ in 10 mL of water at 0 °C was slowly added to the mixture to initiate a reaction for 1 h. With the addition of 0.022 mol NaBF_4,_ while stirring, a suspension was made. The suspension was filtered after cooling to −3 °C. Ice cold water and ether were used to rinse the yielded light-yellow solid. The product was subsequently dried in a vacuum and stored in a desiccator at 4 °C.

*p-Sulfo terminated*: An amount of 2.0 g 4-sulfo aniline was dissolved into 25 mL water solution with 0.58 g Na_2_CO_3_ while heating and stirring. Then, the solution was cooled down to room temperature, and 0.75 g NaNO_2_ was added. A solution made of 2.5 mL concentrated hydrochloric acid and 16 mL of water was made in advance and cooled in an ice bath before adding to the sulfo aniline-containing solution while stirring. The suspension yielded was filtered to yield white solid. Ice cold water was used to rinse the yield product, followed by vacuum drying and storage in a desiccator at 4 °C.

***Solution preparation:**** Water solution:* An amount of 29.2 mg NaCl (EMD Chemicals Inc.) was added into 10 mL DI water to make 10 mM NaCl water solution. *MeOH solution*: 29.2 mg of NaCl was added to 10 mL HPLC grade MeOH obtained from Fisher Scientific with extended vortex mixing. IPA solution was made in the same fashion, while NaCl was added to saturation. *Water–alcohol Mixture*: water–alcohol mixture was made by mixing prepared water and alcohol solutions. The mixing ratio varies between tests. *Peroxide–alcohol mixture*: 30% (9.77 M) hydrogen peroxide aqueous solution (EMD Millipore) was slowly added into dry MeOH to the required mixing ratio. NaCl was added with gentle stirring.

***Permeance test***: Permeance tests of GO membrane were performed in a flow cell setup made with chemically inert tubes (Tygon 2375, 3/16 inch o.d., 1/16 inch i.d.) and chambers. Tubes of the flow cell setup were secured along a ruler as a reference to liquid flow. The flow cell chamber was machined from solid Delrin (polyoxymethylene-based) block and tapped with tread for tube fittings (Delrin and polypropylene) and screws assembly. Silicone rubber gaskets were used to seal and secure the GO membrane in the flow cell setup. Compressed Ar gas was used to provide constant pressure (10 psi). Solvents were injected and exchanged slowly by syringe before applying pressure on both sides of the stabilized GO membrane. Between tests, the flow cell was carefully rinsed with the solution to be tested by syringe to eliminate the residue solution from previous tests. The flow cell system’s liquid flow was monitored by a video camera (Dino-lite digital microscope) and recorded by VideoVelocity for a time-lapse video. Pictures were taken from the time-lapse video with a same time interval and analyzed with *ImageJ* to identify the velocity of the liquid front moving along the tube. Then the transmembrane flux is calculated, as shown in the following equation:Flux = A_tube_ × v_liquid_ ÷ A_membrane_(1)
where A_tube_ is the cross-section area of tube used, the v_liquid_ is the velocity of liquid front calculated from the distance liquid front moved during the set time interval, and the A_membrane_ is the surface area of the membrane defined by the opening of the Delrin support. The calculated flux was presented in the unit of liter/m^2^·h or LMH. The permeance of the solution was then calculated by the following equation, as shown:Permeance = Flux ÷ P(2)
where P is the pressure applied by a compressed Ar tank as mentioned above. The permeance was presented in a commonly used unit of liter/m^2^·h·bar, or LMH/bar.

***Osmosis test****:* The osmotic behavior of the GO membrane was examined to study the water selectivity of the membrane. Osmosis tests were performed with mechanically secured GO membrane and custom-made glass U-tubes (Figure 1B), a short straight tube (i.d. 4.2 mm), and a longer U-shaped tube (i.d. 9.3 mm). After the GO membrane was secured with spring clamps between glass tubes and sealed by o-rings, different solutions were filled into glass tubes on each side of the GO membrane to equal heights without air bubbles. Then, both tubes were covered with parafilm to minimize the evaporation of the solutions. Between tests, the setup and membrane were rinsed with DI water to remove residue solution from previous tests. The osmotic flow was monitored and recorded using the same method mentioned in the previous section. The osmotic flux was calculated using Equation (1) by the same method described above. The result osmotic flux was presented in LMH. ***Average permeance and flux calculation:*** The permeance measured by pressure-driven tests and measured osmotic flux shows a substantial decrease during the test period. (Figure 2 and Figure 3 scatter plot). Due to this decrease, an average value during the sampling time interval is not representative of the GO membrane’s best performance. To calculate initial permeance and osmotic flux, data collected within the initial 10 h was modeled with a linear fit:Y = AX + B(3)
where Y represents the pressure-driven permeance or osmotic flux, X represents test time, A is a negative constant revealing the decay rate of permeance or flux over time under the best fitting condition, and the intercept of the y-axis, B, giving the initial value of permeance or osmotic flux. The average and standard deviation of such initial values of permeance and osmotic flux from different tests were calculated and plotted in a column chart format. (Figure 2 and Figure 3 column chart).

***The X-ray diffraction (XRD):*** The X-ray diffraction (XRD) patterns of GO membranes were obtained on a Rigaku desktop X-ray diffractometer using Cu Kα (1.54059 Å) radiation with the X-ray generator operating at 20 kV and 30 mA. Data were collected for a 2θ range of 5.0° to 20.0° at an angular resolution of 0.01°/s. The deconvolution of diffraction peaks were conducted by multipoint nonlinear curve fir (Lorentz) in origin. Then d-space was calculated using Bragg’s law.

***Fourier-Transform Infrared (FT-IR) spectra:*** FT-IR spectra were recorded over a range from 400 cm^−1^ to 4000 cm^−1^ with a resolution of 4.0 cm^−1^ using a point ATR mode of Nicolet iS10 FT-IR spectrometer (Thermo Fisher Scientific, Waltham, MA, USA). A piece of GO membrane was soaked in 10 mM NaCl water solution for about 10 min then removed from solution for measurement. The same GO membrane was re-soaked in 10 mM NaCl water–alcohol mixture solution for about 10 min and recorded spectra using the same technique.

## 3. Results and Discussion

There is a wide variety of reported permeability values reported in the literature, largely due to membrane fabrication and testing methods. The reported permeability of water ranges from 0.007 to 33 µm × LMH/bar, as shown in Table 1. Generally, higher fluxes or permeances are seen for thinner membranes, which are more prone to defects. GO membrane should exceed a critical thickness, which is dominated by its fabrication process, for example, at least 8 nm for a filtered membrane, to be considered as defect-free theoretically [12,22]. Aside from the thickness of the membrane, the microstructure of the GO membrane is another dictating factor of the membrane performance [23]. The graphene oxide membrane used in this study was fabricated through a slip-cast process with an optimal shear rate of non-Newtonian suspension results in highly oriented GO flakes compared to membranes made by common filtration or drop-casting process [25]. It is important to note that these GO membranes were unstable with pure solvents and required 10 mM (mmol/L) salt added, presumably to screen the charge of anionic oxide groups on the surface of GO flakes. For example, within 3 h, the membranes can swell from their initial thickness of 5 µm to about 1 mm when immersed in pure water. With extended time, the GO eventually redispersed into the solution. Such a phenomenon is consistent with previous reports and attributes to the hydrophilicity of graphene oxide [15]. It is also important to note that the handling of GO membranes while mounting in a flow cell or changing solvents can introduce defects and artificially induce high permeance. Hence a procedure to mount GO membranes between metal wire mesh and introduction of solvents via syringe through rubber gasket was developed to have consistent flux measurements. Direct blue 71 added to feed solutions in pressure-driven permeance tests was not found in permeate solution, demonstrating the integrity of GO membrane was maintained during experiments.

The average initial permeance of single solvents H_2_O, MeOH, and IPA (10 mM NaCl as GO stabilizer) were found to be 0.379 ± 0.153, 0.334 ± 0.160 and 0.417 ± 0.308 LMH/bar. It is important to note that the average initial permeance is defined by values of the y-axis intercept of a linear fit of data from corresponding tests. Thus, the value of the average initial permeance is expected to be higher than the observed values shown in the scatter plot. Considering the reported permeance of solvents was measured differently with membranes of different thickness from different groups, it is essential to normalize the permeance into permeability, which was defined as “membrane thickness × permeance” for practical comparisons. With our 5 µm thickness, the corresponding average permeability of water, MeOH and IPA are 1.90 ± 0.77, 1.67 ± 0.80, and 2.09 ± 1.54 µm × LMH/bar. The reported water permeability values in Table 1 have a large range of 0.007 to 33 µm × LMH/bar, with our slip-cast GO membranes in the higher range of permeance performance. It is important to note that this is a relatively thick membrane avoiding defects that can dominate the observed permeance of thinner membranes. Thus we consider this is a reliable report of permeability through GO membranes of ordered microstructure via slip-casting. It is also important to note that dry membrane thickness is used in the permeance calculation compared to the literature reports. This value can be adjusted proportionately by the observed d-spacing swelling described later in this report but is not tabulated here.

However, the permeance of water–alcohol mixtures was significantly lowered by a factor of 5–10, consistent with pervaporation results where the permeation flux dropped from 1.36 kg/m^2^h with pure water to 0.3 kg/m^2^h with 90 wt % EtOH [25]. Each set of dots from the scatter plot in Figure 2 shows the permeation drop trends on the same membrane in a continuous (~20 h) test. Between tests, solutions were exchanged via syringe to avoid mechanically disturbing the membranes or inducing defects. Upon addition of water–alcohol mixtures, the permeance dropped 10-fold (from over 0.15 LMH/bar to ~0.03 LMH/bar in the presented test) over 8 h for IPA and more rapidly with MeOH (quickly stabilized at ~0.02 LMH/bar in the presented test). Importantly, the fluxes could be recovered when switched back to pure water, indicating a reversible change in the membrane took place. For instance, in the shown test, the reduced flux of water–IPA immediately recovered from ~0.03 LMH/bar to over 0.15 LMH/bar. The observed permeance of water-MeOH stabilized ~0.02 LMH/bar, which was lower than the observed permeance of water–IPA stabilized at ~0.03 LMH/bar (Figure 2 scatter plot). This difference in decreased permeability is likely due to the difference in molecular sizes of the alcohols. Differences in mixture viscosity, surface tension, and other factors may also contribute to the difference in permeability decrease but cannot be quantified here. Normally, after replacing the water–alcohol mixed with water (10 mM NaCl stabilizer), the permeance would recover to its initial level immediately. However, after exchanging from the Water-MeOH mixture to water, the permeance stabilized at ~0.05 LMH/bar rather than the observed initial water permeance value of over 0.15 LHM/Bar in this test. Because the MeOH is smaller in molecular size and has a higher molar concentration (12.4 M, or mol/L) than IPA (6.5 M) in the corresponding water–alcohol mixture, it is expected that MeOH molecules enter the GO interlayer more readily compared to the IPA. In addition, it was reported that MeOH has a higher affinity to GO compared to water and other alcohols [21]. Therefore, the ability to restore original water flux was reduced after MeOH exposure, presumably due to residue MeOH trapped in GO interlayer space. The average initial permeance value also agrees with the phenomenon where the water–alcohol mixture induced a reversible decay of permeance. The average initial permeance of water-MeOH and water–IPA was found to be 0.086 ± 0.007 and 0.160 ± 0.081 LMH/bar, which is lower than the initial permeance of water at 0.379 ± 0.153 LMH/bar (Figure 2 column chart). This can be explained by considering an assumed reversible reaction on the GO membrane that is not instantaneous, while the calculated average initial permeance was defined by the permeance at starting moment of tests. Even though the water permeance failed to recover to its initial value after extended exposure to alcohols, the recovery of permeance is still significant, indicating the reduction of water permeance is reversible.

Ion rejection by GO membranes is demonstrated by the osmotic flux 0.31 LMH observed with 10 mM NaCl and 1 M (mol/L) NaCl water solution opposite the membrane. The flux decreased about 20% from 0.31 LMH to 0.25 LMH over 20 h period (Figure 3 test A scatter plot). Since the osmotic pressure is directly proportional to the concentration of draw solute, a 20% decrease should correspond to a 20% change in draw solute concentration, which would require the addition of 0.175 mL solvent to the draw solution (0.7 mL initial). However, the volume increase of the draw solution was found to be about 0.24 mL, which should cause more than a 20% drop in flux. Likely, the decrease in flux was induced by a localized high concentration of solution near the GO membrane at the permeate side. The maintained osmotic flux (pressure) over time is direct evidence of ion rejection and the GO membrane’s water selectivity. The average of initial osmotic flux was found to be 0.338 ± 0.087 LMH (Figure 3, test A column).

Interestingly, osmotic flux was also observed with 1:1 water–alcohol mixture being a draw solution and water being feed solution (both with 10 mM NaCl added), as shown in Figure 3 test B. This is a clear indication of selectivity of water over water–alcohol mixture. The high value of initial osmotic flux, 1.263 LMH and 0.637 LMH for water-MeOH and water–IPA drawing solution, at the beginning of the test is caused by higher molarity of solute (alcohol) in draw solution at 50 vol% (12.4 M and 6.5 M for MeOH and IPA). This is surprising since the permeance of pure IPA was found comparable to the permeance of pure water. This indicates that, in the presence of water, GO membranes form selective water channels excluding alcohol flow, whereas the GO is permeable to pure alcohols. This is the first observation of GO membranes becoming water selective in the presence of water–alcohol mixtures in an osmotic setup.

To make a more direct comparison to test A in Figure 3, test C in the same figure has water–alcohol mixtures on both sides and has a 0.99 M salt concentration difference across the membrane. The key distinction is that with alcohol in the feed solution, we expect a flux decrease similar to what was observed from pressure-driven permeance experiments (Figure 2). Consistent with our hypothesis, the osmotic flow was found to be less than 0.1 LMH at first and dropped to less than 0.02 LMH over the course of 20 h. (Figure 3 test C scatter plot) The averaged value of *initial* osmotic flux was 0.106 and 0.074 LMH for the case of MeOH and IPA being in the solvent. (Figure 3 test C column) This is significantly lower than the flux found under the conditions of Figure 3 test A, 0.338 ± 0.087 LMH, where no alcohol was presented in the feed solution. Presumably, with alcohol on the feed side, the osmotic flow would draw alcohol into the GO membrane. However, in the case of alcohol only on the draw side, alcohol was driven away from the membrane by osmotic flux. When the solution pair of 10 mM and 1 M NaCl in 1:1 water–alcohol (test C) was changed back to 10 mM NaCl in water, and 1:1 water–alcohol (repeat of test B), the osmotic flux recovered to its initial value at about 0.55 LMH, as shown in the scatter plot for repeated test B in Figure 3. The recovered average initial flux was found to be 0.788 LMH for MeOH and 0.747 LMH for IPA. Such recovery of osmotic flux (from 0.1 LMH level to greater than 0.7 LMH) further confirmed the reversible nature of the osmotic flux reduction phenomenon induced by water–alcohol mixtures chemically reducing GO membranes.

Chemical modification of the GO membrane was performed to prevent flux decrease caused by water–alcohol mixtures. The hypothesis was that by having high charge density on GO surfaces and GO nanochannels’ entrances, water would be favored over the less polar and more sterically hindered solvents like IPA. Concentrated sulfuric acid, 4-sulfo benzene diazonium, and direct blue 71 (coupled with 4-carboxyl benzene diazonium) were grafted onto the GO surface by the method used in prior work with CNT membrane [46]. Among the chemical modification tested, 4-sulfo benzene diazonium most significantly promoted osmotic flux of water (to 0.90 LHM from 0.27 LMH of pristine GO membrane), possibly due to stronger ion rejection. The contact angles were 72.5°, 70.0°, 73.3° concerning pristine, concentrated sulfuric acid-treated and 4-sulfo benzene diazonium treated GO membranes, indicating modest differences in hydrophilicity after treatment. However, when the feed solution had alcohol present, a dramatic drop in osmotic flux was seen (Figure 4 test A, test C). This indicates the modification of GO charge density is insufficient to counter the alcohol-induced hindrance of water transport. Figure 4 further shows the consistency of the alcohol interference phenomena with different surface chemistries. Therefore, the decreased trans-membrane flux, induced by the presence of water–alcohol in the feed side of the GO membrane, is not simply due to surface charge or hydrophobicity effects.

To study GO membranes’ structure, we used X-ray diffraction (XRD) to monitor the spacing change between 2-D planes of GO treated with different solvents and mixtures as shown in Figure 5A. Air-dried GO membrane has a d-spacing of 8.515 Å, which is comparable with previous studies, with a sharp XRD peak indicating a well-packed laminated structure of GO [12,14,25]. As expected, the d-spacing increased with the intercalation of large solvent molecules from 12.188 Å. 13.262 Å and 16.202 Å when the membrane was immersed in water, MeOH and IPA, respectively. The expansion in d-space is in good agreement with the increasing size of intercalated molecules.

However, the mixture of water–IPA, as shown in Figure 5B, gave an unexpected decrease in d-spacing compared to the GO-water case. Upon the addition of IPA, a shoulder (at ~9.9 Å) is seen, and the main peak gradually shrinking from 12.875 Å to 9.927 Å during the 100 min period. This behavior is very different from the simple intercalation of molecules, such as IPA, which alone increased the d-spacing to 16.202 Å. Similarly, it is not likely that the cluster of alcohol–water molecules would enter the interlayer space of the GO membrane as it would cause expansion of the d-space rather than observed shrinkage. Interestingly, contraction in d-space is about 2 Å, which is similar to the size of removed–COOH groups for a GO reduction reaction. A similar d-space decrease with an alcohol–water mixture solvent was also observed in another study [19]. However, different from our observation, they found alcohol molecules induce less expansion in GO d-spacing compared to water when intercalated in their GO membrane. Nevertheless, a decrease in GO d-spacing is expected to decrease trans-membrane flux.

Another important experimental observation is that the color of the GO membrane changed from brown to black by the addition of alcohol into water-wetted GO membranes Figure 5C. This suggests a partial reduction reaction of GO to r–GO with the presence of water and alcohol. The initial brown color can be recovered by drying the membrane (not shown). Similar reduction/oxidation reaction induced color change of GO was also reported by another study [18].

To investigate chemical reaction with water–alcohol mixtures, infrared spectra with corresponding XRD were done, as shown in Figure 6. Pristine GO membranes (black curve) show characteristic peaks at 1725 cm^−1^ (C=O), 1619 cm^−1^ (C=C), and a very broad O-H peak at 3300 cm^−1^. With the addition of water (blue) and water–alcohol (red) mixture, the C=O peak at 1725 cm^−1^ disappeared. This reversible result is likely due to the formation of hemiacetals or hydrates (gem-diols) of carbonyl moieties in the presence of water or alcohol. Those are typically unstable so that they are completely reversible. The formation of hemiacetals or hydrates typically needs a negligible amount of acid or base catalyst [45,49,50]. Furthermore, of note is that the O-H stretch at 3200 cm^−1^ sharpens and increases to 3300 cm^−1^ for both water and mixture, indicating a more ordered system. In the case of water–alcohol mixtures, small peaks at 2900 cm^−1^ of C-H stretch and multiples 1500–900 cm^−1^ indicate the incorporation of alcohol. The IR spectra return to the same as pristine GO with drying with C=O peak returning.

The corresponding XRD with solvent treatment is shown in Figure 6B,D. The GO d-spacing increases with the addition of water (from 8.598 Å to over 13.18 Å), as expected. With the addition of MeOH, there is a small decrease in d-spacing to 12.65 Å, but the formation of a shoulder peak at 10.54 Å. A shoulder would indicate a mixture of sample states, compared to Figure 6 shifting of the entire peak. The deconvoluted peaks show, with the addition of MeOH and IPA, the d-space of regions of the GO membrane decreased ~2.1 Å and ~2.4 Å accordingly. Therefore, the d-space change cannot be attributed to the molecular size of a larger solvent but a reaction that removes chemical groups on the GO, particularly reducing carboxylate groups. Interestingly, GO has recently been found to be catalytically active for the oxidation of alcohols [39,45,51].

The proposed mechanism shown in Figure 7A involves a catalytic cycle that oxidizes alcohol to its aldehyde and carboxylic acid form and causing a reduction of GO at the same time [39,45,51]. Once reduced, GO has less or no reaction activity and contracts in d-spacing with loss of carboxylates. This, in turn, dramatically slows the permeation of solvents. Hypothetically, once the reduced GO membrane was re-oxidized, the d-spacing can be recovered, restoring its reactivity and solvent permeability. This would be consistent with the observation of the reversibility of flow phenomena observed during drying. However, the flow cell and the osmosis test tubes were sealed from the air to prevent the evaporation of the solution. Thus, with limited oxidant from the air and excessive alcohol as reductant, the GO membrane remained at a reduced state, and the flux of solvent was limited by the reduced d-space. To test the catalytic cycle hypothesis, stoichiometric amounts of oxidant were added to the mixture with the expectation to recover flow rates. Figure 7B shows water permeance at approximately 0.5 LMH/bar. With the addition of 10% MeOH (*v/v*) in water, the permeance dropped to approximately 0.16 LMH/bar, which is about 1/3 of the value of water permeance. Subsequently, H_2_O_2_ was introduced to restore the oxidation state and d-space of the GO membrane. With the addition of 10% MeOH (2.47 M)—90% H_2_O_2_ (8.79 M) mixture and proper compensation of bubble generated during reaction, the permeance of mixture solution recovered to approximately 0.6 LMH/bar. The recovery of flux with added oxidant is in a good alignment to our hypothesis and purposed catalytic reaction. With abundant oxidant supply, the water–alcohol-reduced rGO was oxidized to its initial state. Thus, the d-space of the GO membrane was recovered, allowing the initial fast transmembrane flux. Similarly, potassium permanganate was also tested as an oxidant added to the alcohol–water mixture and yielded similar results.

## 4. Conclusions

With proper stabilization methods, both chemically and mechanically, shear-aligned graphene oxide-based membranes were able to show steady and repeatable performance. The permeance of water, MeOH and IPA was observed to be 0.379 ± 0.153, 0.334 ± 0.160, and 0.417 ± 0.308 LMH/bar, respectively. Considering the membrane used here has a higher thickness (~5 µm) compared to membranes used by other researchers (commonly 10–50 nm), the permeance value was normalized with the thickness of the membrane yielding water permeability of 1.32 µm × LMH/bar. Knowing that various studies of GO membranes showing the water permeability range in a wide spectrum of 7 × 10^−2^ to 3.3 × 10^1^ µm × LMH/bar as the fabrication methods of membranes vary (Table 1), the water permeability performance of our GO membrane lays in the high end of the spectrum, thanks to aligned GO grains. We believe this observation is accurate since dye permeation control experiments excluded any significant role of defects larger than 2 nm. A similar slip-casted GO membrane with precise control of the fabrication process from Majumders’ group showed a slightly better performance of 10.7 µm × LMH/bar, but with a lower thickness of 150 nm [24]. This indicates, with proper optimization, the permeance of our current GO-based membrane can be improved while maintaining defect-free 2-D nanochannels.

Although individual solvents show fast permeance through the GO membrane, a dramatic flux reduction over a magnitude (to 1/10 of its initial strength) was observed when the GO was approached by a mixture of water–alcohol solution. Note that the reduced permeance of the mixture solution is lower than the permeance of any individual of the component of the solution. This indicates the reduction of the flux was not caused by a simple single “slow” molecule species but through a more complicated mechanism. In addition, the permeance of a single solvent solution can be recovered upon removal of the mixture solution, which indicates a non-permanent effect on the GO membrane.

Osmosis experiments provided more insight into the liquid transportation behavior within the GO membranes. The osmotic flow between water solutions of different salt concentrations indicated the GO membrane’s nanochannels are water selective and can effectively reject salt ions. When testing with draw/feed pair of water–alcohol mixture/water, the osmotic flow was observed as well. Knowing that previous tests show that alcohols by themselves can permeate through the GO membrane, such phenomenon indicates that the GO membrane favors water over alcohol in a water–alcohol mixture and forms selective water channels. Consistent with the pressure-driven permeance tests, reduction of osmotic flux was also observed when water–alcohol mixtures are introduced into the feed side of the GO membrane during osmosis tests. Surface chemical modifications were made in an attempt to alter the selectivity of the molecule at the entrance of the GO membrane in the hope that polarized groups on the surface will ease the entrance energy barrier of the water molecule, therefore, eliminate the flux reduction with mixture solution. A significant increase of osmotic flux (to about 4 times of initial value after treatment with 4-sulfo benzene diazonium) was observed, but the phenomena of dramatic flux reduction with water–alcohol mixtures was still observed, suggesting a different mechanism other than chemical selectivity at the GO plane entrances.

The observation of color change of the GO membrane leads to the hypothesis of reduction of the GO, which was supported by FTIR characterization. In the water–alcohol mixture, the C=O peak disappeared than GO at the dry state or dried water–alcohol mixture treated GO. XRD data shows that upon the exposure of water–alcohol mixtures, the d-spacing of GO deceased by ~2.2 Å, which is in good agreement with the size of a carboxyl group. Combining these FTIR and XRD data, we were led to believe that the GO was reduced by the water–alcohol mixture reversibly losing its carboxyl groups. As reported by Bielawski and Yang’s group, the GO can catalytically oxidize alcohols at the cost of itself being reduced [39,45,51]. We also found the reduced GO can be re-oxidized to recover its catalytic activity. Our observation of reversible transmembrane flux reduction upon exposure to the water–alcohol mixture can be explained by the proposed catalytic reaction cycle. In the presence of excessive alcohol in an aqueous solution as reductant, the GO membrane was reduced, causing a d-space shortening, which leads to a reduction in transmembrane flux. Once the excessive alcohol reductant was removed, the reduced GO membrane will be re-oxidized by environmental oxidants, such as oxygen in the air. The reduced d-space of the GO membrane was thus expanded, allowing the recovery of transmembrane flux. To further prove the purposed catalytic reaction cycle, H_2_O_2_, as the oxidizer, was added into the MeOH containing mixture solvent and successfully prevented the reduction of permeation flux. This observation supports the mechanism of GO reduction being responsible for the observed GO d-spacing decrease and a significant decrease in permeability of water–alcohol mixtures. In short, the water–alcohol mixture-induced d-space reduction of GO membranes was demonstrated to be reversible and explained by a catalytic GO reduction reaction cycle. The selectivity of water, as well as rejection of alcohol and salt ions, was also demonstrated in mixtures. This insight is needed for the GO membrane’s application in the biofuel production field, where one must carefully control the oxidation environment to ensure stable fluxes and productivity.

## Figures and Tables

**Figure 1 membranes-11-00317-f001:**
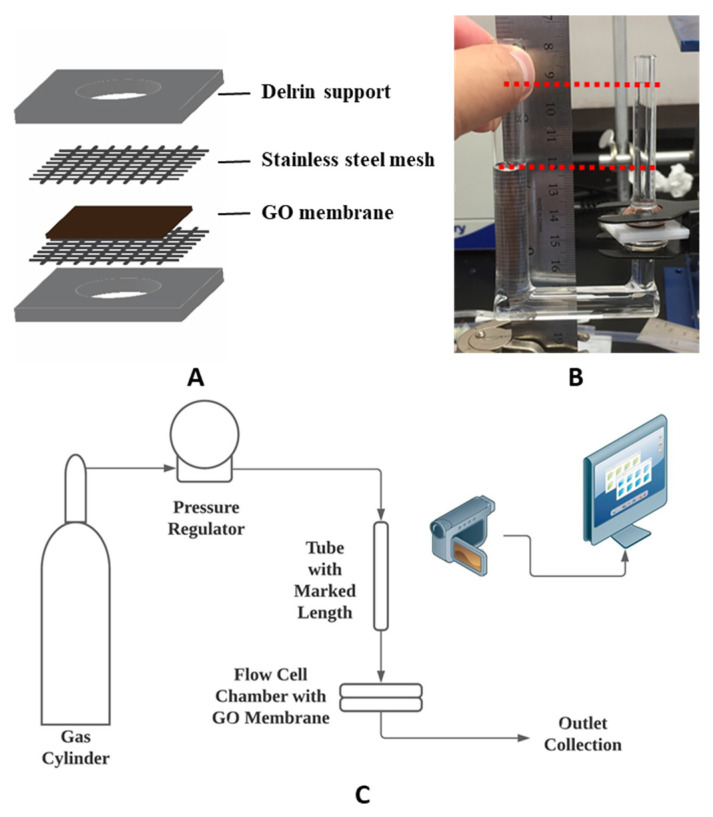
(**A**) Schematic of graphene oxide (GO) membrane supports; (**B**) photograph of u-tube test setup in osmosis test; (**C**) schematic of the pressure-driven permeance test setup.

**Figure 2 membranes-11-00317-f002:**
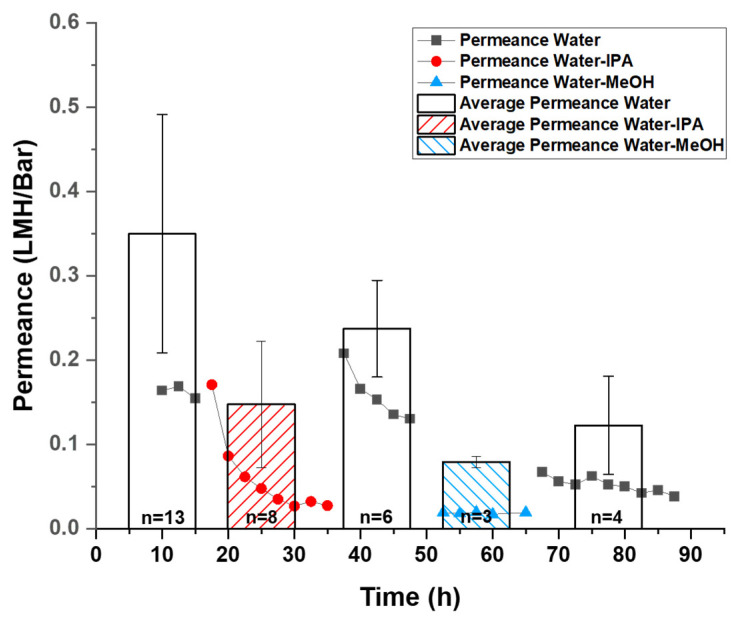
Pressure-driven permeance vs. time trend of the same GO membrane with different solvent mixtures (scatter plot) and average *initial* permeance calculated from tests on different membranes (columns). Water–alcohol (IPA or MeOH) mixtures are 1:1 *v/v*. The pressure applied was 10 psi. Each set of dots from the scatter plot shows the behavior of permeance in one continuous test with the solution mixture indicated by the legend. Between tests, solution mixtures were exchanged via syringes into the flow cell. The average permeance was calculated from “n” repeated tests under the same condition as described in the Experimental section. The number of tests, n, was labeled inside the corresponding columns.

**Figure 3 membranes-11-00317-f003:**
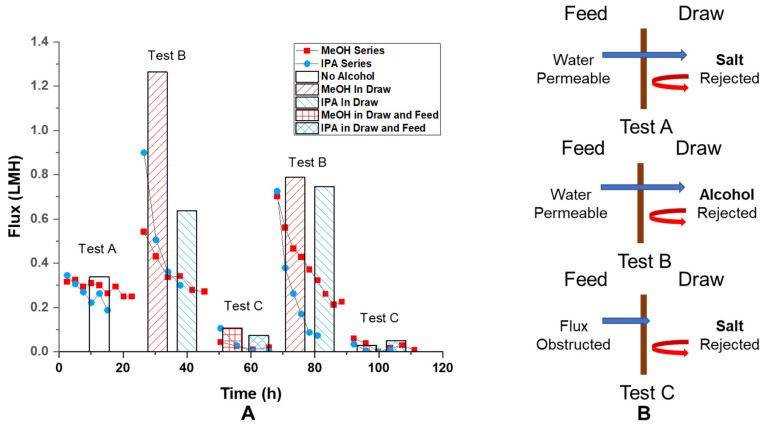
(**A**) Observed osmotic flux vs. time trend on the same GO membrane under different test conditions (scatter plot) and average extrapolated initial flux of different GO membranes under various test conditions (columns); (**B**) schemes of cross-membrane flux under different test conditions. Test A: feed: 10 mM (mmol/L) NaCl in water, draw: 1 M (mol/L) NaCl in water, n = 7; test B: feed: 10 mM NaCl in water, draw: 10 mM NaCl in water–alcohol mixture, n = 1 for MeOH and 4 for IPA at the first test, n = 2 for MeOH and 3 for IPA at the repeat test; test C: feed: 10 mM NaCl in water–alcohol mixture, draw: 1 M NaCl in water–alcohol mixture n = 2 for MeOH and 3 for IPA at the first test, n = 2 for MeOH and 2 for IPA at the repeat test. Test B and C were repeated under the same condition. The number “n” is the number of tests used to calculate the average flux under the same condition specified by the corresponding legend (columns). The water–alcohol mixture is made by mixing 1:1 *v/v* water and alcohol (MeOH or IPA).

**Figure 4 membranes-11-00317-f004:**
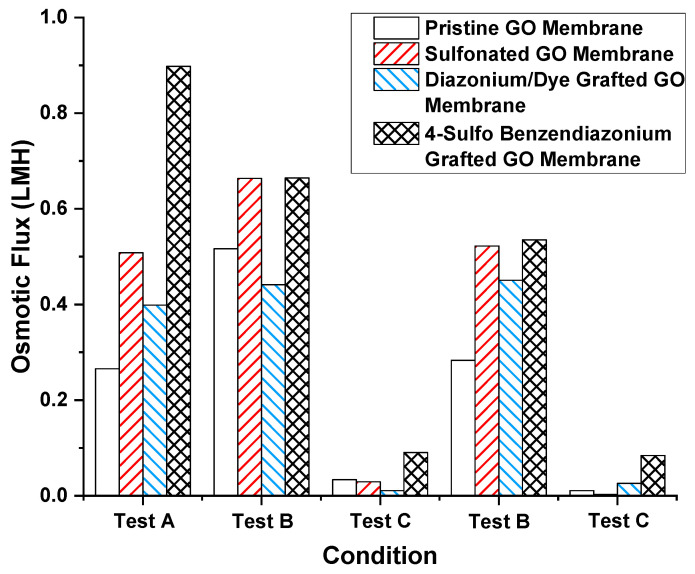
Osmotic flux of different chemical-treated GO membranes under different sequential test conditions. Test A: feed: 10 mM NaCl in water, draw: 1 M NaCl in water; test B: feed: 10 mM NaCl in water, draw: 10 mM NaCl in water–IPA mixture; test C: feed: 10 mM NaCl in water–IPA mixture, draw: 1 M NaCl in water–IPA mixture. The water–IPA mixture is 1:1 *v/v* water and IPA.

**Figure 5 membranes-11-00317-f005:**
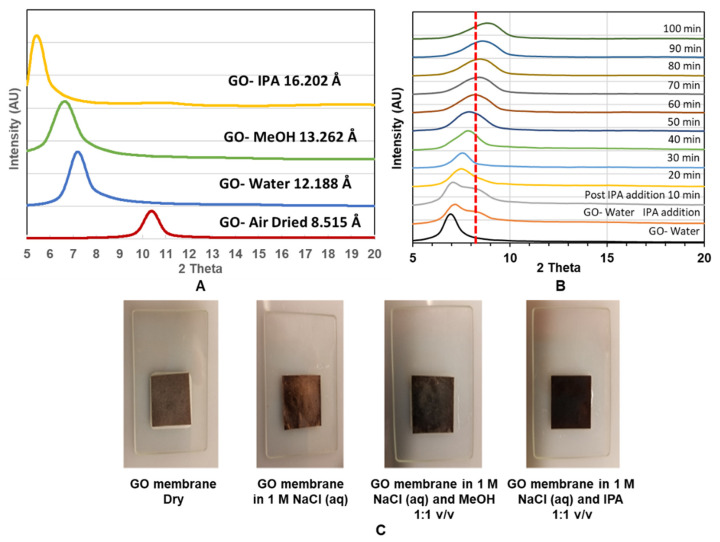
(**A**): XRD of GO immersed in indicated solvents; (**B**): XRD as a function of time for GO immersed in water and peak shift after addition of IPA; (**C**): Color change observed when the GO membrane was immersed in different solutions.

**Figure 6 membranes-11-00317-f006:**
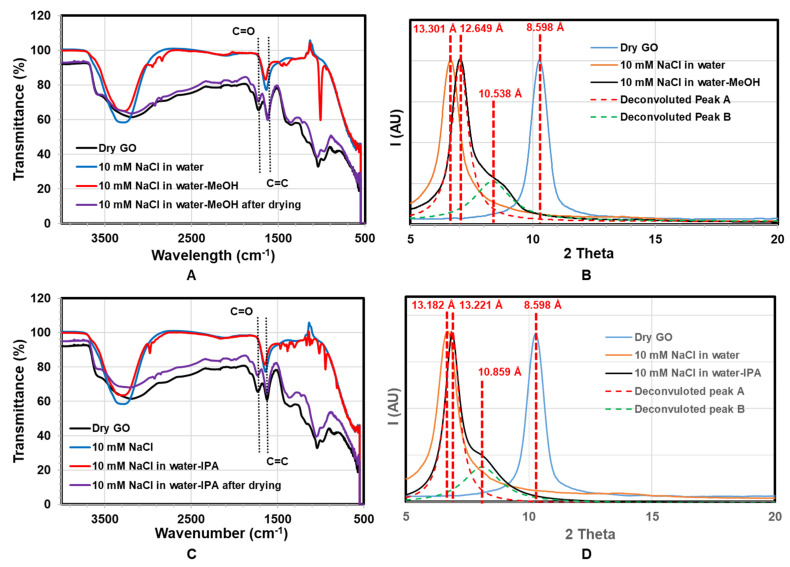
ATR-IR and corresponding XRD data on the same membrane. (**A**) ATR-IR of GO membrane immersed in water and water-MeOH; (**B**) corresponding XRD data of GO membrane immersed in water and water-MeOH; (**C**) ATR-IR of GO membrane immersed in water and water-IPA; (**D**) corresponding XRD data of GO membrane immersed in water and water-IPA.

**Figure 7 membranes-11-00317-f007:**
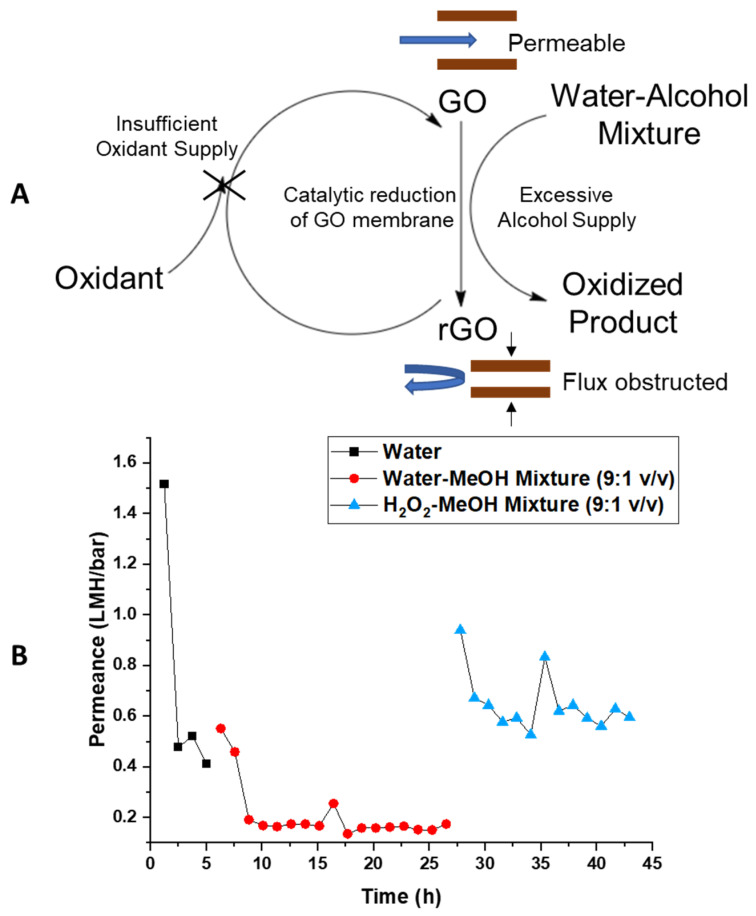
(**A**) Scheme of purposed catalytic reduction of GO membrane by alcohols; (**B**) pressure-driven permeance vs. time trend with added oxidant reversing the catalytic reduction of GO membrane. Figure also shows reduced d-space of GO membrane obstructing cross-membrane flux at reduced state.

**Table 1 membranes-11-00317-t001:** Literature reports of GO membrane permeabilities.

Source	Test Method	Feed Solution	Thickness (µm)	Permeability µm × LMH/bar	Comment
This report	Pressure-driven	Water	5	1.895	The osmotic pressure generated by 1 mol/L NaCl and 10 mmol/L NaCl across the membrane was 49.58 bar
		MeOH		1.67	
		IPA		2.085	
	Osmosis	Water	5	0.034	
[9]	Pressure-driven	Water	5	16–33	
[12]	Pressure-driven	Water	0.15	0.045–0.225	
[13]	Pressure-driven	IPA	0.018	0.432	
		MeOH	0.018	1.355	
		Water	0.018	1.62	
[18]	Pressure-driven	Water	0.016	0.08–0.096	EDA crosslinked
				0.192–0.24	EDA crosslinked + reduction
[22]	Pressure-driven	Water	0.048	0.178	
		IPA	0.048	0.965	
		IPA:water 9:1	0.048	0.077	
[24]	Pressure-driven	Water	0.15	10.65	Slip-casted membrane
			0.17	1.7	Vacuum-filtered membrane
[30]	Pressure-driven	Water	0.07	0.56	
			0.7	19.32	
[33]	Pressure-driven	Water	0.04	0.190–0.485	
[34]	Pressure-driven	Water	0.6	18.9	
[35]	Pressure-driven	Water	0.1	9.2	
		IPA	0.1	13	
		EtOH	0.1	18	
[23]	Pervaporation	BuOH:water 9:1	0.231	0.587	Pressure-filtered membrane
			0.384	0.909	Vacuum-filtered membrane
			0.447	0.412	Evaporation-made membrane
[25]	Pervaporation	Water	5	6.733	
		EtOH:water 9:1	5	1.485	
[28]	Pervaporation	Alcohol:water 85:15	0.5	0.99	
[29]	Pervaporation	EtOH:water 9:1	0.412	0.546	
[47]	Pervaporation	EtOH:water 85:15	2	30.378	
[48]	Pervaporation	IPA:water 7:3	0.231	0.473	30 °C
			0.231	0.956	70 °C
[14]	Osmosis	Water	1	0.008	
[27]	Osmosis	Water	0.28	0.021	
			0.28	0.009	KCl intercalated
			0.55	0.023	
			0.55	0.011	KCl intercalated
			0.75	0.011	
			0.75	0.007	KCl intercalated

## Data Availability

The study did not report any data.

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
