# Peer review of "Catalytic Reduction of Graphene Oxide Membranes and Water Selective Channel Formation in Water–Alcohol Separations"

_membranes, 2021, doi:10.3390/membranes11050317_

Round 1
Reviewer 1 Report
In this manuscript, the authors prepared and modified GO membranes, and executed permeance and osmosis tests with the membranes under different water/alcohol environments. The authors observed that their GO membranes presented high permeability with water/alcohols and were water selective in water-alcohol mixtures. They also studied the drop in permeance of water-alcohol mixtures and attributed the drop to the decrease in GO d-spacing. The authors further proposed a catalytic reduction mechanism of GO membrane with the presence of water and alcohol.
The authors reported some really interesting phenomena with GO membranes including the formation of water selective channels and the reduction of GO in water-alcohol mixture. However, the manuscript can be constructed in a better way to emphasize the most important information in a clearer manner. The current structure is a bit hard to follow and some discussions are sidetracking. For example, the authors provided extensive description on the calculation and comparison of permeance, which is not that closely related to the main topic. The language in some paragraphs, especially in the introduction section, will need to be improved. Please also check for inaccurate expressions and typo.
Here are some general comments/questions:
- Since the manuscript is titled “catalytic reduction of GO membranes…”, the authors should provide relevant background information on GO catalysis and discuss the potential application.
- The authors defined permeability as “membrane thickness * permeance of solvent” in order to compensate for the influence of changing thickness. As the interlayer spacing of GO membranes can change in different solvents, does this affect the thickness of membrane and has this been taken into account for the calculation of permeability?
- In the experimental section, how was the membrane sample sealed in the U-tube test, with an o-ring? It would also be helpful if the authors can provide a photo or diagram of the flow cell set-up.
- How were the membranes characterized after chemical modification?
- The authors proposed a catalytic reduction mechanism of GO in water-alcohol mixture. What role does water play in this process? In the schematic diagram it’s indicated that alcohol can be oxidized by GO – then was the same effect observed in pure alcohol?
Line-specific comments:
Line 72: typo
Line 85: typo
Line 235: Figure 2 looks very busy and is confusing. The scatter plots in gray (permeance of water) do not seem to be representative in terms of both the values and the trend over time (especially from 10 hr to 40 hr). Is there a reason to have dual y-axis in this graph since both are based on the same set of data?
Line 244 – 245: How were the first set of permeance values calculated? Why the average initial permeance of water and IPA are lower? Please explain.
Line 266: What about the influence of surface tension and viscosity?
Line 256 – 281: It would be helpful to plot the recovery of permeance. The numbers here are inconsistent and confusing.
Line 282: Figure 3A, same comments as Figure 2. It’s fine to overlay flux and average flux, but why does it have to have dual y-axis with different tick values? Did the authors also calculate average flux for Test B & C at later time stage?
Line 324 – 326: Was the flux drop phenomenon reversible or not? This can be confusing to readers.
Line 329: What are the two groups of Test B & C in this graph? Please clarify. Also check for typos in the caption.
Line 396: Figure 6D is scratchy.
Line 467: typo
Author Response
"Please see the attachment."

Reviewer 2 Report
This work lacks a complete nomenclature and the definition of the quantities and their detailed calculation formulae.
For example in line 185 – 190: … which should not so approximately be integrated into the text)
Under the heading "Average permeance and flux calculations", for example, there are no formulas with the quantities used and the corresponding dimensions.
Line 198: It should be described here the method exactly and not to point out that the calculation of the osmotic flow takes place with a similar method as for the calculation of the pressure-related permeation flow.
Furthermore, all textual errors and designations in the tables and graphs need to be corrected.
For example, in Figure 2: n is indicated as the number of measurements, but the respective number of measurement points is only applicable to one mixture in this diagram.
Litres are denoted by l and then again by L, which is usually indicated by bar and not by bar, one time it is µ and then again u, what is M,m,H mol etc.?
Mixtures are indicated as water-MeOH or IPA or EtOH, but what is meant by water-alcohol mixtures?
Four terms are described as result variables: Permeance, osmotic flux, permeate flux, permeability, whereby the respective driving forces are missing in many places in the dimensions given.
Confusing and unclear, difficult to understand and compare with other values or data.
There is no equilibrium data, all data are in the non-stationary range and all data is in constant change. How can usable and reproducible results be obtained from this.
Perhaps longer periods of time are required here to condition the membranes in the respective medium. Of course, this is connected with the problem that the compositions in front of and behind the membrane are also constantly changing due to diffusion and an equilibrium is only reached when the concentrations in front of and behind the membrane no longer change. But then the driving force goes towards 0 and the chemical potential is the same on both sides. Here, however, the critical question arises as to whether the experimental concept is suitable for this.
Line 448: Which spectrum is meant here and why is it judged here that this is correct. If values are measured, they must be reproduced accordingly and the methods used for this must also be presented in a comprehensible manner.
Many textual corrections are to be made regarding spacings to and from numbers, dimensions, designations.
Line 467: …reject slat ions ?
Line 471: How can this be explained in a proper way!
Line 489 “The catalytic cycle purposed here are in great alignment with our observations.” What is meant by this and how is it justified?
Self-assessments should give way to factual and scientific presentation and justification.
Then the significant question is what this data can be used for.
The type and nature of the membrane may be interesting, how a use of this active layer is possible and how stable it is over a longer period of time is of course also very essential.
Author Response
"Please see the attachment."

Round 2
Reviewer 1 Report
The authors have addressed my questions/comments, and the manuscript has been improved significantly. However, I'd like to point out that Figure 2 has not been updated as stated below.
"Line 235: Figure 2 looks very busy and is confusing. The scatter plots in gray (permeance of water) do not seem to be representative in terms of both the values and the trend over time (especially from 10 hr to 40 hr). Is there a reason to have dual y-axis in this graph since both are based on the same set of data?
Reply: We agree it is a complicated figure and we had several versions to convey the several concepts, settling on this one. The scatter plot was included to demonstrate the decaying trend of measured permeability during tests, which is important for the reader to see. Although the x-axis was plotted in a fashion with continuous time as shown, it is important to note that between different sets of data (line connected dots or columns) the sample flow cell was depressurized for solution exchange then pressurized again for testing as mentioned in the experimental section. This brings up the necessity to trace back the initial value of permeance at the moment when solution exchange and pressurization were done. The calculated initial values from tests were then averaged from different runs and plotted as the column chart.
To simplify, we removed the dual y-axis (with a different scales) as suggested. Hopefully, now it is better presented."
Author Response
"Please see the attachment."

Reviewer 2 Report
The permeance of the pure solvents water, methanol and isopropanol all are in the same range with deviations of +/- 50 % and more, as this is shown as a result. When water is added to the alcohols, this permeability drops by a factor of 40 to 50. This may be interesting, as noted, but where is the advantage here? In processing, it is always mixtures from which water is to be removed.
What are the results of this study and where can progress be seen and what do these new results mean and what can be concluded from them?
An overview in a separate section of all the designations, symbols and their dimensions used is necessary here for consistent designation and clarification.
Author Response
"Please see the attachment."

Round 3
Reviewer 2 Report
Again spacings (only examples)
line 213: liter/m2 –h ?
Spacing in line 422: shrinking from 12.875Å to 9.927Å and can this be determined up to 1/1000 Å?
line 452: 3200cm-1
line 474: Fig.7 A
line 486: Fig.7 B
line 518: …. If it was not caused by a simple single "slow" molecular species, but by a more complicated mechanism, how can this flux reduction be explained? How can this complicated mechanism be described?
Author Response
"Please see the attachment."
